# Effect of Exogenous Melatonin on Performance and Mastitis in Dairy Cows

**DOI:** 10.3390/vetsci11090431

**Published:** 2024-09-13

**Authors:** Yunmeng Li, Zhiqiang Cheng, Wenting Ma, Yaqi Qiu, Tuo Liu, Bingyu Nan, Mengfei Li, Long Sun, Wentao Liu, Haina Yin, Caidie Wang, Xiaobin Li, Changjiang Zang

**Affiliations:** 1Xinjiang Laboratory of Herbivore Nutrition for Meat and Milk, College of Animal Science, Xinjiang Agricultural University, Urumqi 830052, China; m1762691038@163.com (Y.L.); cheng07162022@163.com (Z.C.); m2175367682@126.com (W.M.); liutuo20240808@163.com (T.L.); nannanyouya163@163.com (B.N.); lmf0212@126.com (M.L.); muxiangmu163@163.com (L.S.); 19199277197@163.com (H.Y.); wcd@xjau.edu.cn (C.W.); lixb@xjau.edu.cn (X.L.); 2Karamay Green City Agricultural Development Co., Karamay 834000, China; m18599169310@163.com; 3Xinjiang Urumqi Rural Revitalization Guidance Service Center, Urumqi 830000, China; 15739433358@163.com

**Keywords:** lactating dairy cows, melatonin, mastitis, performance

## Abstract

**Simple Summary:**

Mastitis in dairy cows is a major problem for livestock production. In this study, we investigated the effects of exogenous melatonin on the milk somatic cell count and yield and the serum anti-inflammatory factors, antioxidant indices, and melatonin in dairy cows. Exogenous melatonin was shown to increase the milk component content, reduce the somatic cell count, and improve the antioxidant capacity and immune responses.

**Abstract:**

Mastitis is an important factor affecting the health of cows that leads to elevated somatic cell counts in milk, which can seriously affect milk quality and result in huge economic losses for the livestock industry. Therefore, the aim of this trial was to investigate the effect of melatonin on performance and mastitis in dairy cows. Forty-eight Holstein cows with a similar body weight (470 ± 10 kg), parity (2.75 ± 1.23), number of lactation days (143 ± 43 days), BCS (3.0–3.5), milk yield (36.80 ± 4.18 kg), and somatic cell count (300,000–500,000 cells/mL) were selected and randomly divided into four groups: control (CON group), trial Ⅰ (T80 group), trial Ⅱ (T120 group), and trial Ⅲ (T160 group). Twelve cows in trial groups I, II, and III were pre-dispensed 80, 120, and 160 mg of melatonin in edible glutinous rice capsules along with the basal ration, respectively, while the control group was fed an empty glutinous rice capsule along with the ration. The trial period was 37 days, which included a 7-day adaptive phase followed by a 30-day experimental period. At the end of the trial period, feeding was ended and the cows were observed for 7 days. Milk samples were collected on days 0, 7, 14, 21, 28, and 37 to determine the somatic cell number and milk composition. Blood samples were collected on days 0, 15, 30, and 37 of the trial to determine the serum biochemical indicators, antioxidant and immune indicators, and the amount of melatonin in the blood. The results showed that the somatic cell counts of lactating cows in the CON group were lower than those in the T120 group on days 14 (*p* < 0.05) and 28 (*p* < 0.01) at 1 week after melatonin cessation. The milk protein percentage and milk fat percentage of cows in the T120 group were higher than those in the CON group (*p* < 0.01). The total protein and globulin content in the T120 group were higher than those in the CON group (*p* < 0.01). In terms of antioxidant capacity and immunity, the cows 1 week after melatonin cessation showed higher superoxide dismutase activity and interleukin-10 contents (*p* < 0.01) compared with the CON group and lower malondialdehyde and tumor necrosis factor-alpha contents (*p* < 0.01) compared with the T120 group. The melatonin content in the T120 group was increased relative to that in the other groups. In conclusion, exogenous melatonin can increase the content of milk components, reduce the somatic cell count, and improve the antioxidant capacity and immune responses to a certain extent. Under the experimental conditions, 120 mg/day melatonin is recommended for mid- to late-lactation cows.

## 1. Introduction

Mastitis is a common disease in modern farming and levies considerable costs to the dairy industry worldwide, including treatment costs, the loss of production, and milk stoppage costs [1]. Mastitis affects almost all lactating mammals, especially high-producing cows [2,3]. Mastitis not only decreases milk production in dairy cows, but also increases the number of somatic cells in milk, which can lead to a decrease in milk quality and negatively impact economic efficiency. In China, the annual loss due to mastitis in dairy cows is estimated at approximately CNY 135 million; thus, this disease causes huge economic losses to the dairy industry and severely restricts its development [4]. Therefore, reducing the incidence of mastitis in high-yield dairy cows through nutritional modification strategies is important for improving animal husbandry.

Melatonin is an antioxidant that was first discovered in 1958 by Lerner et al. [5] in bovine pineal glands. Subsequently, it was detected in other species (bacteria, fungi, animals, and plants) [6,7,8]. Melatonin is produced not only in the pineal gland, but also in other organs and tissues, such as the gastrointestinal tract, brain, kidneys, and liver [9], and it has been well documented as an effective free radical scavenger and antioxidant [10]. A previous study found that the somatic cell count (SCC) and cortisol concentrations in the milk of cows with mastitis were significantly reduced following a subcutaneous injection of melatonin [11]. Jiménez et al. [12] showed that the subcutaneous implantation of melatonin in the broad ear of goats at the beginning of lactation reduced the SCC in goat milk, which could improve milk quality. Melatonin has anti-inflammatory, antioxidant, sleep-promoting, mood-improving, and reproduction-regulating properties [13]. Huang et al. [14] subcutaneously injected melatonin to treat a post-viral infection in mice and found that the treatment decreased the expression of tumor necrosis factor α (TNF-α), interleukin 6 (IL-6), and interferon γ (IFN-γ) and increased the production of interleukin 10 (IL-10) and transforming growth factor β (TGF-β). The results of our group’s previous studies on a lipopolysaccharide (LPS)-induced mouse mastitis model and the preventive effects of melatonin on mastitis in mice showed that melatonin administered via gavage significantly suppressed the serum levels of pro-inflammatory factors (TNF-α, interleukin 1β (IL-1β), and IL-6), decreased the levels of malondialdehyde (MDA), and increased the serum levels of anti-inflammatory factors (IL-10 superoxide dismutase (SOD), glutathione peroxidase (GSH-Px), and catalase (CAT)), which ameliorated the mammary tissue damage caused by LPS [15].

Therefore, this study aimed to determine the effects of exogenous melatonin on the SCCs and mastitis in cows during mid- and late lactation. The results provide insights into the application of melatonin for improving the practical production of high-yield dairy cows.

## 2. Materials and Methods

### 2.1. Ethics Statement

All procedures involving animal care and management were approved by the Institutional Animal Care and Use Committee of the Xinjiang Agricultural University (Urumqi, China).

### 2.2. Experimental Design, Diet, and Management

This study was conducted at the Xinjiang Changji Jiyuan Herding Co. (Changji, China). Forty-eight Holstein cows with a similar body weight (470 ± 10 kg), parity (2.75 ± 1.23), number of lactation days (143 ± 43 days), BCS (3.0–3.5), milk yield (36.80 ± 4.18 kg), and somatic cell count (300,000–500,000/mL) were selected for this study and randomly assigned to four treatment groups: control (CON), trial group I (T80 group), trial group II (T120 group), and trial group III (T160 group). Twelve cows were assigned to each group, and edible glutinous rice capsules (length, 21 mm; diameter, 7 mm) containing 0, 80, 120, or 160 mg/day of melatonin (Anhui Zesheng Technology Co., Ltd., Hefei, China) per cow were provided along with the basal ration. The trial period was 37 days, which included a 7-day adaptive phase followed by a 30-day experimental period. At the end of the trial period, the melatonin treatments were stopped and the animals were observed for 7 days (Figure 1).

During the trial period, all the trial animals were kept under the same feeding and management conditions and were provided with food and water ad libitum. Manure was removed regularly every day to keep the enclosures clean. Milk samples were collected three times a day at 06:30, 14:30, and 22:30. After collecting the milk samples, the cows were fed the TMR diet regularly at 07:00, 15:00, and 23:00 each day. The melatonin capsules were provided at 08:00 each day according to the Chinese Dairy Cattle Feeding Standard (NY/T 34-2004) [16]. The composition and nutritional levels of the basal rations are shown in Table 1.

### 2.3. Sample Collection

At the beginning of the experimental period, the morning, midday, and evening TMR feeds were collected and mixed once every 10 days; then, 600 g samples were collected by the tetrad method and stored in the refrigerator at −20 °C. These samples were used to determine the routine nutrient composition of the feed.

Milk production was recorded every 7 days during the experimental period and statistically analyzed.

Three milk samples were collected at 06:30, 14:30, and 22:30 on days 0, 7, 14, 21, 28, and 37. The milk samples collected in the morning, midday, and evening were mixed into 100 mL portions at a ratio of 4:3:3 and stored at −20 °C to determine the routine nutrients (milk fat percentage, milk lactose percentage, milk protein percentage, and non-milk-fat solids content) and the SCC.

On days 0, 15, 30, and 37, 15 mL blood samples were collected before the morning feeding from the tail root of all cows using common blood-collection tubes (without anticoagulant). The serum was separated by centrifugation at 3500× *g* for 15 min at 4 °C, dispensed into 1.5 mL Eppendorf tubes, and stored at −20 °C. The remaining serum was used to determine the blood melatonin levels and antioxidant and anti-inflammatory indices.

### 2.4. Sample Testing and Analysis

Feed samples and leftovers were subjected to dry matter (DM, 105 °C), organic matter (OM, No. 922.06), crude protein (CP, No. 988.05), crude fat (EE, No. 922.06), calcium (Ca, No. 977.29), and phosphorus (P, No. 995.11) analyses according to AOAC procedures (AOAC, 2000) [17]. Neutral and acid detergent fibers were determined using the method described by Van Soest et al. [18].

The milk fat percentage, lactose percentage, milk protein percentage, non-fat solids, and ash content were determined using a milk composition analyzer (Bulgaria Lactoscan Model MCC-50, Nanjing Oxi Science and Trade Co., Nanjing, China). The SCC was determined using a somatic cell counter (LactoScan SCC; Henset Biotechnology Co., Ltd., Shanghai, China).

A fully automated blood biochemistry analyzer (Beckman AU5800; Beckman Coulter Ltd., Brea, CA, USA) was used to analyze the serum biochemical indices, including glucose (GLU), albumin (ALB), globulin (GLB), total protein (TP), urea nitrogen (UN), alanine aminotransferase (ALT), azelaic aminotransferase (AST), total bilirubin (TB), alkaline phosphatase (ALP), triglyceride (TG), total cholesterol (CHO), high-density lipoprotein (HDL), and low-density lipoprotein (LDL).

The serum inflammatory factors TNF-α, IL-1β, IL-6, and IL-10 were determined using a kit (Beijing Huaying Institute of Biotechnology, Beijing, China) according to the manufacturer’s instructions. The serum levels of the antioxidant factors SOD, GSH-Px, CAT, and MDA were determined by a colorimetric method.

### 2.5. Statistical Analyses

One week after the cessation of melatonin supplementation (trial day 37), the SCC, milk yield, milk composition, antioxidant indices, anti-inflammatory indices, and serum melatonin content were analyzed by a one-way ANOVA using SPSS 19.0 (IBM Corp., Armonk, NY, USA), and multiple comparisons were performed using Duncan’s method. The results were expressed as the mean, and the degree of variation was expressed as the standard error of variance. A value of *p* < 0.05 indicated a significant difference.

Data analyses of the milk yield, milk composition, antioxidant indicators, anti-inflammatory indicators, and serum melatonin content at the time of melatonin feeding were performed using the PROC MIXED method (SAS 9.4, SAS Institute Inc., Cary, NC, USA), which included the time points, fixed effects of treatments and their interactions, and random effects. The model is as follows:Y_ijk_ = μ + trt_i_ + cow_j_ + day_k_ + (trt_i_ × day_k_) + e_ijk_
where Y_ijk_ is a measure of the dependent variable of the j-th cow of the i-th treatment on the k-th day, μ is the overall mean of the dependent variable trt, trt_i_ is the fixed effect of the i-th treatment on the dependent variable, day_k_ is the fixed effect of the k-th day of treatment on the dependent variable trt, trt_i_ × day_k_ is the fixed effect of the interaction between trt_i_ (T80, T120, and T160) and day_k_ (time: trial days 0, 7, 14, 21, and 28) on the dependent variable, and e_ijk_ is the random residual effect of the j-th cow in the i-th treatment on the k-th day. The data are presented as the least-squares mean and the mean standard error, and *p* < 0.05 indicates a significant difference.

## 3. Results

### 3.1. Effect of Feeding Melatonin and Cessation of Feeding for 1 Week on Somatic Cell Counts in Dairy Cows

The effects of melatonin supplementation and melatonin cessation for one week on the SCCs of cows are shown in Figure 2. Compared with that of the CON group, the SCC of cows was reduced in the T120 group on day 14 (*p* < 0.05), in the T120 and T160 groups on day 28 (*p* < 0.01), and in the T80 group (*p* < 0.05). In addition, 1 week after melatonin cessation, the SCCs of the cows in the trial groups were lower than that in the CON group on day 37 of the experiment, although the differences were not significant (*p* > 0.05).

### 3.2. Effects of Feeding Melatonin and Effects 1 Week after Melatonin Cessation on the Performance of Dairy Cows

The effects of feeding a melatonin supplementation and the effects 1 week after melatonin cessation on dairy cow production are shown in Table 2. The T120 group showed a higher milk fat percentage (*p* < 0.05) and higher milk fat yields, milk protein, and milk protein yields compared with the CON group (*p* < 0.01). This group also had a higher milk protein and milk protein yield compared with the T80 group (*p* < 0.01) and higher values compared with the T160 group (*p* < 0.05).

Compared to those in the CON group, lactating cows in the trial groups showed an elevated trend in all the indicators except for non-fat milk solids and ash, although treatment, linear, and quadratic differences were not observed (*p* > 0.05).

### 3.3. Effects of Feeding Melatonin and Effects 1 Week after Feeding Cessation on Serum Antioxidant Indices in Dairy Cows

The effects of melatonin supplementation on the serum antioxidant indices in dairy cows are shown in Table 3. Compared with that in the CON group, the SOD activity was higher in the T120 and T160 groups (*p* < 0.01) and the GSH-Px activity was higher in the T120 group (*p* < 0.05). The MDA content was lower in the T120 and T160 groups (*p* < 0.01) and significantly lower in the T80 group than in the CON group (*p* < 0.05).

The effect of melatonin on the serum antioxidant indices in dairy cows 1 week after feeding cessation is shown in Table 3. The SOD, GSH-Px, and CAT activities were higher and the MDA content was lower in all the experimental groups than in the CON group; however, the linear and quadratic differences were not significant (*p* > 0.05).

### 3.4. Effects of Feeding Melatonin and Effects 1 Week after Feeding Cessation on Serum Inflammatory Indices in Dairy Cows

The effects of melatonin supplementation and the effects 1 week after feeding cessation on the serum inflammatory indices in dairy cows are shown in Table 4. The TNF-α concentration was lower in the T120 group than in the CON and T80 groups (*p* < 0.01). The IL-6 concentration was lower in the T120 and T160 groups than in the CON group (*p* < 0.05).

Compared with those in the CON group, the TNF-α and IL-6 concentrations decreased and the IL-10 concentrations increased in all the trial groups, although significant linear and quadratic differences were not observed (*p* > 0.05).

### 3.5. Effects of Feeding Melatonin Supplementation and Effects 1 Week after Feeding Cessation on Serum Level Melatonin in Dairy Cows

The serum melatonin levels were higher in the T120 group than in the CON group (*p* < 0.01), although the serum melatonin levels did not differ between the trial groups (*p* > 0.05). However, the serum levels were higher in the T80 and T160 groups than in the CON group (*p* < 0.05) (Table 5).

After the cessation of melatonin supplementation for 1 week (day 37 of the trial), significant differences were not observed (*p* > 0.05) in the melatonin content between the trial and control groups.

## 4. Discussion

### 4.1. Effect of Exogenous Melatonin on Somatic Cell Number in Dairy Cows

Mastitis in dairy cows is a frequently encountered problem in animal husbandry; it is mainly characterized by an elevated milk SCC and reduced milk quality [19,20], and is often used as an important indicator for evaluating the udder health of dairy cows. The higher the SCC, the poorer the milk quality [21], and cows are diagnosed with subclinical mastitis when the milk SCC exceeds 300,000/mL. In this study, the SCCs of cows decreased significantly during melatonin supplementation compared with 1 week after cessation, with a sustained decrease in the SCCs of cows on days 0, 14, and 28. The most significant decrease in the SCCs occurred in the T120 group, which may be closely related to the antioxidant and anti-inflammatory activities of melatonin. As a mitochondrial antioxidant, melatonin plays an important role in scavenging reactive oxygen species (ROS), inhibiting the oxidative stress that occurs in mastitis, and activating anti-inflammatory effects that inhibit LPS-binding-associated protein expression and reduce the release of pro-inflammatory factors in the mammary epithelial cells of cows. Yao et al. [22] supplied 40 mg/day or 80 mg/day of melatonin to the rumen of dairy cows by the “rumen bypass melatonin feeding method”, and they observed a significant decrease in the SCCs of the cows and the effective alleviation of mastitis symptoms.

### 4.2. Effect of Exogenous Melatonin on Milk Yield and Milk Composition of Dairy Cows

Melatonin was first isolated from the pineal glands of dairy cows and investigated to determine its role in the physiological reproduction of seasonally reproducing animals [23]. Subsequent studies have widely applied melatonin to regulate the reproductive performance of mice, sheep, and dairy cows. In addition, the use of melatonin has a significant effect on mastitis in dairy cows by reducing the SCCs, improving milk production and milk quality, and better serving people. With the improvement of people’s living standards and consumption levels, increased attention has been paid to the quality of milk, which is rich in protein, fat, vitamins, lactose, and mineral elements [24]. Moreover, the composition of cow milk is close to that of human milk and widely accepted by most cultures. Infant formula is usually based on cow milk supplemented with appropriate compounds. Therefore, milk quality is important for the health of infants and adults, particularly for the growth and development of infants [25].

Milk production and composition have a direct impact on the economic efficiency of dairy farms and represent important indicators of the health status of cows during lactation. The milk composition, milk protein, milk fat, lactose, and non-fat solids are key determinants of milk quality. Milk proteins are synthesized from amino acids in the mammary glands [26], while lactose is produced by lactose synthase in the mammary glands and represents the main source of energy [27]. In the present study, although significant changes in milk yield were not observed during melatonin supplementation compared with 1 week after stopping melatonin, a significant change was observed in the milk composition, with varying increases in the milk fat rate, milk protein rate, milk fat yield, and milk protein yield and decreases in the lactose rate. Lactating cows function better under high levels of melatonin, which activates a large number of mammary epithelial cells, enhances their metabolic activity, improves lactation, and increases milk production [28]. This is consistent with the findings of Molik et al. [29], who showed that melatonin supplementation in ewes significantly increased the milk fat and milk protein content and improved milk quality. However, Auldist et al. [30] performed melatonin supplementation in late-lactating cows and showed that it reduced the milk yield and lactose concentration while increasing the milk fat and milk protein content.

### 4.3. Effect of Exogenous Melatonin on Antioxidant Indexes in Dairy Cows

Oxidative damage is a noteworthy problem in animal production because it affects enzyme viability and gene expression, and ultimately disrupts the redox balance of cells [31]. The dynamic balance between the generation and scavenging capacity of ROS is closely related to cellular activity and animal health [32]. During the lactation stage of dairy cows, a large number of free radicals are generated during oxidative metabolism, and if they are not scavenged in time, the accumulated free radicals in the body will destroy the structure and function of the cells, which become prone to oxidative stress. Oxidative stress impairs the antioxidant status by increasing lipid peroxidation and decreasing plasma antioxidant concentrations. All these reactions affect the animal’s health and decrease the animal’s production performance. GSH-Px, SOD, MDA, and CAT can be used to evaluate the physiological health of animals. GSH-Px and SOD play a key role in inhibiting lipid peroxidation and preventing macromolecular damage. In this study, the serum SOD and GSH-Px activity and the MDA content were significantly altered after melatonin supplementation and remained altered after the melatonin treatments were ceased for 1 week. Moreover, the GSH-Px and SOD activity was significantly or very significantly higher in the T120 group than in the CON group during melatonin supplementation, while the MDA content was very significantly reduced. One week after the melatonin treatments were ceased, the GSH-Px and SOD activity and MDA content did not show significant differences among the groups. These findings indicate that melatonin played an obvious role in antioxidant action, prevented peroxidation by regulating and removing peroxidation-generated free radicals, and transformed active substances into inactive substances. This is consistent with the findings of Choudhary et al. [33], who showed that the exogenous addition of melatonin to Chhotanagpuri ewes increased the serum SOD and CAT activity and reduced the MDA content.

### 4.4. Effect of Exogenous Melatonin on Inflammatory Response in Dairy Cows

Increases in pro-inflammatory factors (e.g., TNF-α, IL-1β, IL-2, and IL-6) in lactating cows lead to inflammation and compromise the intestinal integrity, resulting in impaired liver functions [34]. TNF-α, IL-1β, and IL-6 are the major pro-inflammatory cytokines, and they play a crucial role in cell survival and apoptosis, with elevated levels of these pro-inflammatory factors triggering inflammation [35,36]. In contrast, IL-10 is mainly secreted by macrophages and T cells. As an anti-inflammatory factor, IL-10 selectively blocks the synthesis of pro-inflammatory factors and protects the body from inflammation-induced damage [37]. In this study, the pro-inflammatory factors in the serum of cows supplemented with melatonin were reduced compared with those 1 week after melatonin supplementation ceased. In particular, the TNF-α and IL-6 contents were significantly reduced and the IL-10 content was increased in the T120 group, which suggests that feeding melatonin can reduce the abundance of matrix metalloproteinases [38], inhibit the expression of pro-inflammatory factors by regulating macrophages via increasing anti-inflammatory factors, and reduce the oxidative stress in the organism, thus reducing the inflammatory response in cows. This is consistent with the findings of Yao et al. [22]. Supplementing dairy cows with melatonin led to significantly reduced serum levels of TNF-α and IL-6 and increased levels of IL-10.

### 4.5. Effect of Exogenous Melatonin on Serum Melatonin in Dairy Cows

The concentration of melatonin in the blood is related to the season, light, animal’s physiological state, half-life, and feed intake, with seasonal factors being the main factors affecting the concentration [39,40]. In the non-breeding season, melatonin can be used to regulate the estrous time of ewes so that they enter estrus early [41]. Moreover, melatonin can reduce the mortality rate of newborns while promoting the growth and development of the fetus [42]. In addition to seasonal factors, melatonin is regulated by circadian rhythms. In vivo, tryptophan, a precursor substance of melatonin, is hydroxylated to 5-hydroxytryptophan by tryptophan hydroxylase [43], and after decarboxylation, it becomes 5-hydroxytryptophan. Serotonin increases during the day, and the activity of aryl alkylamine acetyltransferase in the pineal gland increases 70–100-fold at night [44]. Finally, acetyl-5-hydroxytryptamine methyltransferase converts N-acetyl-5-hydroxytryptonin to melatonin. In this study, the melatonin concentration in the blood of dairy cows was increased by exogenous dietary melatonin supplementation, which reached the corresponding tissues and organs through blood circulation. This is consistent with the findings of Shi [45], who showed that exogenous melatonin concentrations in the blood of dairy cows can be increased by an intravenous injection, subcutaneous injection, subcutaneous implantation, or oral instillation of exogenous melatonin. In contrast, 1 week after melatonin cessation, significant differences in the blood melatonin concentration were not observed relative to those during supplementation, which is due to the half-life of melatonin being approximately 30 min [46]. Therefore, melatonin withdrawal for 1 week had little effect on the melatonin concentration in the blood of dairy cows.

## 5. Conclusions

Under the trial conditions, the supplementation of 120 mg/day of melatonin for dairy cows can improve their milk composition (milk fat rate, milk protein rate, milk fat rate, lactose yield, and milk protein yield), antioxidant capacity (improved SOD and GSH-Px activity), and immune capacity (increased IL-10 content and reduced TNF-α and IL-6 content); reduce the somatic cell count; and have a certain effect on the prevention and treatment of mastitis. It is recommended to supplement the cow diet with 120 mg/day of melatonin.

## Figures and Tables

**Figure 1 vetsci-11-00431-f001:**
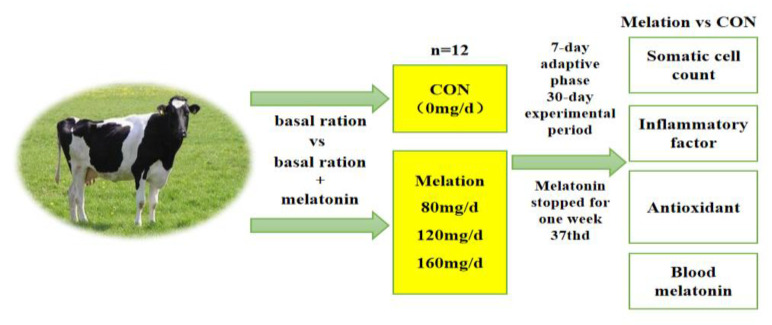
Trial design and grouping diagram.

**Figure 2 vetsci-11-00431-f002:**
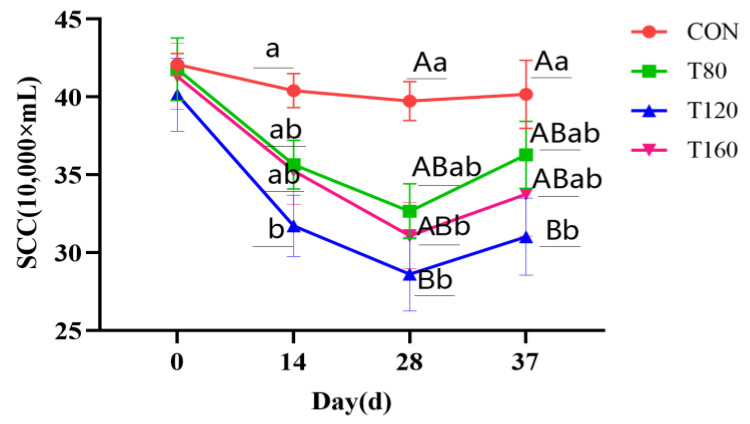
Effect of melatonin on somatic cell count in dairy cows.T80, trial 80 mg/day; T120, trial 120 mg/day; T160, trial 160 mg/day; SEM, standard error of mean; SCC, somatic cell count; CON, control with no MT; T80, T120, and T160, fed basal ration and 80 mg/day, 120 mg/day, or 160 mg/day of MT (Senrise Technology Co., Ltd., Anqing, China) per cow, respectively. In the figure, the same letter indicates no significant difference (*p* > 0.05) and different lowercase or uppercase letters indicate significant differences (*p* < 0.05; *p* < 0.01).

**Table 1 vetsci-11-00431-t001:** Composition and nutrient levels of the basal diet (DM basis).

Ingredients	Content	Nutrient Components ^2^	Content
Alfalfa	15.20	DM	96.42
Corn silage	24.17	OM	92.64
Tablet corn	6.47	CP	15.57
Corn	24.23	EE	3.96
Concentrate 1818 ^1^	17.95	NDF	38.39
Fatty powder	0.75	ADF	16.45
Cottonseed meal	4.04	Ca	0.85
Cottonseed	1.41	P	0.46
Beet granules	4.78	NE_L_/(MJ/kg)	6.60
Methionine	0.06		
Sodium bicarbonate	0.56		
NaCl	0.38		
Total	100.00		

^1^ Concentrated feed 1818 is a commercial feed, and its nutrient content is as follows: dry matter, 86%; crude protein, 29.5%; crude fiber, 12%; crude ash, 18%; calcium, 1.3%; total phosphorus, 0.6%; sodium chloride, 1.3%; and lysine, 0.8%. Per kg, it contains the following: VA, 12,000 IU; VD, 3300 IU; VE, 2200 IU; Cu, 42 mg; Zn, 181 mg; Mn, 53 mg; Fe, 56 mg; I, 1.9 mg; Co, 0.41 mg; and Se, 0.8 mg. ^2^ The net energy for lactation (NE_L_) was calculated, and the other nutrient levels were measured. NE_L_ = 9.29 × fat/kg milk + 5.85 × true protein/kg milk + 3.95 × lactose/kg milk.

**Table 2 vetsci-11-00431-t002:** Effects of feeding melatonin and effects 1 week after melatonin cessation on milk yield and milk composition in dairy cows (*n* = 12).

Trial Period	Items	CON	T80	T120	T160	SEM	*p*-Value		
							Treatment	Time	Treatment × Time
0–30 day Melatonin feeding period	Milk yield (kg/day)	36.33	37.05	37.91	37.32	0.445	0.093	0.794	0.999
	Milk fat (%)	3.62 b	3.79 ab	3.87 a	3.79 ab	0.062	0.034	<0.001	0.793
	Milk fat yield (kg/day)	1.31 Bb	1.40 ABa	1.47 Aa	1.41ABa	0.027	0.001	<0.001	0.814
	Milk lactose (%)	4.69	4.62	4.64	4.67	0.033	0.405	0.729	0.995
	Milk lactose yield (kg/day)	1.71	1.71	1.76	1.74	0.024	0.378	0.95	0.999
	Milk protein (%)	3.16 Bb	3.18 Bb	3.31 Aa	3.20 ABb	0.032	0.006	0.040	0.772
	Milk protein yield (kg/day)	1.15 Bb	1.18 Bb	1.26 Aa	1.19 ABb	0.019	0.001	0.004	0.969
	Non-fat milk solids	8.63	8.51	8.56	8.58	0.062	0.578	0.191	0.994
	Ash content	0.70	0.69	0.69	0.69	0.005	0.342	0.108	0.382
**Trial Period**	**Items**	**CON**	**T80**	**T120**	**T160**	**SEM**	** *p* ** **-Value**		
							**Treatment**	**Linear**	**Quadratic**
31–37 day At 1 week after melatonin cessation	Milk yield (kg/day)	37.19	37.24	37.90	37.56	0.336	0.877	0.571	0.781
	Milk fat (%)	3.73	3.79	3.88	3.71	0.073	0.866	0.951	0.459
	Milk fat yield (kg/day)	1.38	1.41	1.47	1.39	0.027	0.675	0.697	0.326
	Milk lactose (%)	4.73	4.71	4.67	4.66	0.036	0.902	0.458	0.996
	Milk lactose yield (kg/day)	1.76	1.75	1.77	1.75	0.021	0.980	0.975	0.832
	Milk protein (%)	3.15	3.22	3.33	3.27	0.041	0.442	0.189	0.435
	Milk protein yield (kg/day)	1.17	1.20	1.26	1.23	0.020	0.404	0.197	0.405
	Non-fat milk solids	8.61	8.44	8.52	8.57	0.065	0.826	0.962	0.427
	Ash content	0.70	0.69	0.68	0.70	0.005	0.436	0.683	0.120

SEM, standard error of the mean; CON, control with no MT; T80, T120, and T160, fed basal ration and 80 mg/day, 120 mg/day, or 160 mg/day of MT (Senrise Technology Co., Ltd., Anqing, China) per cow, respectively. Sampling times were as follows: days 0, 7, 14, 21, and 28 of the trial and one week after cessation of melatonin feeding (day 37 of the trial). The same letter indicates no significant difference (*p* > 0.05); different lowercase or uppercase letters indicate significant differences (*p* < 0.05, *p* < 0.01).

**Table 3 vetsci-11-00431-t003:** Effects of feeding melatonin and effects 1 week after feeding cessation on serum antioxidant indices in dairy cows (*n* = 8).

Trial Period	Items	CON	T80	T120	T160	SEM	*p*-Value		
							Treatment	Time	Treatment × Time
0–30 day Melatonin feeding period	SOD (U/mL)	50.58 Bb	53.00 ABab	55.08 Aa	54.18 Aa	0.871	0.003	<0.001	0.470
	GSH-Px (U/mL)	336.63 b	343.30 b	361.09 a	350.17 ab	5.851	0.028	0.140	0.545
	CAT (U/mL)	32.40	32.68	34.08	32.81	0.671	0.306	0.009	0.488
	MDA (U/mL)	3.92 Aa	3.57 ABb	3.39 Bb	3.48 Bb	0.094	0.001	0.001	0.968
**Trial Period**	**Items**	**CON**	**T80**	**T120**	**T160**	**SEM**	** *p* ** **-Value**		
							**Treatment**	**Linear**	**Quadratic**
31–37 day At 1 week after Melatonin cessation	SOD (U/mL)	50.81	51.69	56.32	53.66	0.933	0.160	0.111	0.332
	GSH-Px (U/mL)	330.24	339.20	363.50	350.52	7.126	0.398	0.192	0.447
	CAT (U/mL)	31.68	33.00	34.73	33.42	0.811	0.636	0.358	0.433
	MDA (U/mL)	4.05	3.51	3.48	3.55	0.098	0.122	0.077	0.111

SEM, standard error of mean; SOD, superoxide dismutase; GSH-Px, glutathione peroxidase; CAT, catalase; MDA, malondialdehyde; CON, control with no MT; T80, T120, and T160, fed basal ration and 80 mg/day, 120 mg/day, or 160 mg/day of MT (Senrise Technology Co., Ltd., Anqing, China) per cow, respectively. Sampling times were as follows: days 0, 15, and 30 of the trial and one week after cessation of melatonin feeding (day 37 of the trial). The same letter indicates no significant difference (*p* > 0.05); different lowercase or uppercase letters indicate significant differences (*p* < 0.05, *p* < 0.01).

**Table 4 vetsci-11-00431-t004:** Effects of feeding melatonin and effects 1 week after feeding cessation on serum inflammatory indices in dairy cows (*n* = 8).

Trial Period	Items	CON	T80	T120	T160	SEM	*p*-Value		
							Treatment	Time	Treatment × Time
0–30 day Melatonin feeding period	TNF-α (pg/mL)	40.02 Aa	39.41 Aa	36.60 Bb	38.39 ABab	0.671	0.003	<0.001	0.072
	IL-1β (pg/mL)	20.64	20.08	19.02	19.91	0.558	0.235	<0.001	<0.001
	IL-6 (pg/mL)	120.59 a	118.34 ab	115.08 b	116.89 b	1.175	0.011	<0.001	0.212
	IL-10 (pg/mL)	11.06 Bb	11.63 ABb	12.55 Aa	11.72 ABab	0.298	0.007	<0.001	0.390
**Trial Period**	**Items**	**CON**	**T80**	**T120**	**T160**	**SEM**	** *p* ** **-Value**		
							**Treatment**	**Linear**	**Quadratic**
31–37 day At 1 week after melatonin cessation	TNF-α (pg/mL)	38.70	37.95	36.25	36.86	0.772	0.699	0.316	0.671
	IL-1β (pg/mL)	20.20	21.20	19.73	20.10	0.680	0.897	0.780	0.827
	IL-6 (pg/mL)	119.75	117.92	111.72	113.59	4.001	0.896	0.513	0.826
	IL-10 (pg/mL)	12.60	11.09	13.07	12.03	0.344	0.208	0.923	0.731

SEM, standard error of mean; TNF-α, tumor necrosis factor alpha; IL-1β, interleukin 1β; IL-6, interleukin 6; IL-10, interleukin 10; CON, control with no MT; T80, T120, and T160, fed basal ration and 80 mg/day, 120 mg/day, or 160 mg/day of MT (Senrise Technology Co., Ltd., Anqing, China) per cow, respectively. Sampling times were as follows: days 0, 15, and 30 of the trial and one week after cessation of melatonin feeding (day 37 of the trial). The same letter indicates no significant difference (*p* > 0.05); different lowercase or uppercase letters indicate significant differences (*p* < 0.05, *p* < 0.01).

**Table 5 vetsci-11-00431-t005:** Effects of feeding melatonin supplementation and effects 1 week after feeding cessation on serum melatonin levels in dairy cows (*n* = 8).

Trial Period	Items	CON	T80	T120	T160	SEM	*p*-Value		
							Treatment	Time	Treatment × Time
0–30 day Melatonin feeding period	MT (pg/mL)	50.42 Bb	53.39 ABa	55.36 Aa	54.44 ABa	1.041	0.008	0.019	0.906
**Trial Period**	**Items**	**CON**	**T80**	**T120**	**T160**	**SEM**	** *p* ** **-Value**		
							**Treatment**	**Linear**	**Quadratic**
31–37 day At 1 week after melatonin cessation	MT (pg/mL)	50.08	53.21	54.68	54.47	1.008	0.358	0.114	0.412

SEM, standard error of mean; MT, melatonin; CON, control with no MT; T80, T120, and T160, fed basal ration and 80 mg/day, 120 mg/day, or 160 mg/day of MT (Senrise Technology Co., Ltd., Anqing, China) per cow, respectively. Sampling times were as follows: days 0, 15, and 30 of the trial and one week after cessation of melatonin feeding (day 37 of the trial). The same letter indicates no significant difference (*p* > 0.05); different lowercase or uppercase letters indicate significant differences (*p* < 0.05, *p* < 0.01).

## Data Availability

All data for this study are available from the corresponding authors.

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
