# Peer review of "Effect of Exogenous Melatonin on Performance and Mastitis in Dairy Cows"

_vetsci, 2024, doi:10.3390/vetsci11090431_

Round 1

Reviewer 1 Report

Comments and Suggestions for Authors

See attchement

Author Response

Draft (vetsci-3170360)

24-August-2024

Statement on the Revision of (vetsci-3170360)

Based on Reviews’ Resports

Author: Zhiqiang Cheng   Correspondence: Changjiang Zang

August 24, 2024

Reply to the Comments of Reviewer 

Reply: We thank the reviewers for comprehensive assessment of our work. We have gone through all the comments in detail and have tried our best to revise our manuscript according to the comments points by point. Additionally, revised portion are marked in red in the revised manuscript. Thanks a lot for your reviewing.

Reviewer 1

The present manuscript provides data from an experiment with a sufficient number of lactating dairy cows referring to the effect of exogenous, orally supplied melatonin. Compared to a control group (no melatonin) the remaining cows were divided in 3 groups defined by the daily dose of melatonin: 80, 120 and 160 mg. The treatment period took 30 days followed by a 7 day of washout period. Recorded were: Cow’s performance including somatic cell count in milk, common biochemical parameters in blood plasma and indicators for oxidative stress. The authors conclude that the oral application of melatonin to lactating cows by a daily dose of 120 mg/head is beneficial regarding the risk to be affected by mastitis.

Reply: We sincerely thank the editor and all reviewers for their professional comments, which we use to improve the quality of the manuscript. The reviewers' comments are listed below in blue markers, and specific questions have been numbered. Revised portions are marked in red in the revised manuscript.

Comments:

  1. Format
  2. The paper needs a strict language related revision.
  3. There are several misleading/useless terms e.g. row 65, “mastitis cow”; row 22, “litter size”; row 37, “extremely significantly higher” A revision regarding terminology and precise wording is required

a.The paper needs a strict language related revision.

Reply: We tried our best to improve the manuscript and made some changes to the manuscript.These change will not influence the content and framework of the paper. In addition, we invite a native English speaker help us polish our article. And here we did not list the changes but marked in red in the revised paper.We appreciated for Reviewers’ warm work earnestly and hope that the correction will meet with approval.

  1. There are several misleading/useless terms e.g. row 65, “mastitis cow”; row 22, “litter size”; row 37, “extremely significantly higher” A revision regarding terminology and precise wording is required

Reply: Thanks for your valuable advice.

Row 65: Revised the “A previous study found a significant reduction in SCC in mastitis cows by means of subcutaneous injection of melatonin and a reduction in cortisol concentrations” to “A previous study found a significant reduction SCC in the milk of cows with mastitis by means of subcutaneous injection of melatonin and a reduction in cortisol concentrations”.

Row 22:The terminology used here is inappropriate. we want to express is parity, which has been revised in the manuscript.

Row 37: Revised the“The total protein and globulin contents of T120 group were extremely significance higher than those of the control group”to “The total protein and globulin contents of T120 group were higher than the control group”.

  1. Abstract
  2. The recorded parameters should be men

Reply: Milk samples were collected on days 0, 7, 14, 28 and 37 for the determination of somatic cell number and milk composition. Blood samples were collected on days 0, 15, 30 and 37 of the trial to determine serum biochemical indicators, antioxidant and immune indicators, and the amount of melatonin in the blood.

3.Introduction

row 52, 53: “.. Mastitis causes .. increase in somatic cell resulting in impairement of ..immune function“. This statement is not correct or at least misleading imprecise and should be reconsidered.

Reply:Mastitis not only causes a decrease in milk production in dairy cows, on the other hand, an increase in the number of somatic cells in milk can lead to a decrease in milk quality, which affects economic efficiency.

4.Methods

  1. Litter size likely means number of lactation; please clear this

Reply: The terminology used here is inappropriate. we want to express is parity, which has been revised in the manuscript.

b.The following data are missed

i.Body mass

Reply: Body weight : 470±10 kg.

  1. Body condition score

Reply: BCS (Body condition) : 3.0-3.5.

iii. Day of lactation

Reply: The cows selected were mid to late lactation cows, lactating for 143 ± 43 days.

  1. The description of the trial (row 100-105) is confusing; please revise

Reply: The selected cows were randomly divided into four groups of 12 cows each, which were control group, trial â… , â…¡ and â…¢, the control group cows were fed with TMR ration + empty glutinous rice capsule (without melatonin), and each cow in trial â… , â…¡ and â…¢ groups was added with 80 mg/d, 120 mg/d and 160 mg/d melatonin per day on the basis of the TMR ration, and melatonin fed to the trial cows was packed in empty glutinous rice capsules.

  1. Tab. 1: Dimensions should be added
  2. Composition, % as fed or % of DM ?

Reply: In Table 1, the composition is the percentage of each nutrient calculated on a dry matter basis.

  1. Nutrients, % DM

Reply: In Table 1, the dry matter content is 96.42%. It has been added in Table 1.

iii. What are: “fatty Powder”, Beet granules”?

Reply:

Fat powder: Fat powder is made by combining different fats and oils according to the ideal fatty acid model and fatty acid composition characteristics of cows at different physiological stages, and then coated or adsorbed with special materials after emulsification and homogenization process. It is a feed additive that can replace conventional fats and oils to provide high energy and functional fatty acid nutrition.

The fat powder has the advantages of good fluidity, easy transportation, storage and use, not easy to oxidize, high digestive efficiency, and fully meet the energy and fatty acid nutritional needs of dairy cows.

Beet granules: Beet granules are the solid parts left over from sugar beets after extracting sugar juice, which are dried and crushed to form granules, which are high in protein and rich in amino acids.

  1. Vit A, Vit E, Fe, Zn, Se should be added at least those quantities which are contributed by the concentrate

Reply:Thank you for your professional advice. We rechecked the concentrate content. and additions and modifications.

  1. Which model is used to calculate NE

Reply:According to the NRC 2001 standard:

NEL=9.29×fat/kg milk+5.85×true protein/kg milk+3.95×lactose/kg milk

  1. Data about the size of capsules should be added

Reply:The information on the capsules is as follows: length 21mm, diameter 7mm

  1. Row 132; what is the meaning of “..into 100 L”

Reply: Three milk samples were collected at three time points (06:30, 14:30 and 22:30) before feeding TMR diet, and a total of 100 mL of milk samples were collected according to the actual milk yield at three time points in the morning, noon and evening according to the ratio of 4:3:3.

  1. Row 133; terminology needs to be reconsidered > milk fat rate, lactose rate, milk protein rate.

Reply: There is a lack of precision in the terminology used here. We have corrected inappropriate terminology in the manuscript. revise milk fat rate to milk fat percentage,  revise lactose rate to milk lactose percentage, revise milk protein rate to milk protein percentage.

  1. The term “positive trial period” is useless, please reconsider the description

Reply: We've made changes here in the manuscript. Modified to adaptive.

  1. Centrifugation: instead of “r/min” “g for x min” should be preferred.

Reply: Based on your professional advice, we have revised the manuscript and carried out the conversion of units.

The radius of the centrifuge is: 6 cm

g=1.1×10-5×6×75002≈3713 g

  1. Row 137; usually, if an anticoagulant is used – here heparin – the harvested fraction after centrifugation is plasma.

Reply:We verified and determined that the blood collection tubes used to collect the blood were ordinary blood collection tubes (without anticoagulant)

  1. It is not clear whether the statistical analysis follow the procedure of an analysis of variance for repeated measurements or a different data processing.

Reply:The statistical methodology here follows repeated measures, with parallel samples for each sample, in order to ensure the accuracy of the data.

  1. What is the standard error of variance? > SEM usually Standard error od the mean

Reply:The manuscript is inaccurate in its representation of standard error of the mean. we want to express the Standard Error of the Mean. We have revised the terminology for Standard Error of the Mean.

  1. Please, define how significant differences are marked (aB etc)

Reply:In the figure, the same letter indicated no significant difference (p>0.05), different lowercase or uppercase letters indicated significant difference (p<0.05; p<0.01), and a, b, A and B indicated significant differences.

5 Results

  1. In general
  2. A detailed description of results regarding the degree of significance should be avoided; the main aspect is a statisticallyconfirmed response

Reply:Based on your professional advice, we have redescribed the results in the manuscript.

  1. I recommend to reconsider the sequence of description of results by “following the route of melatonin”: 1. Melatonin in blood in response in supplementation, 2. Related responses observed by biochemical parameters, e.g. SOD, 2. Milk yield and milk composition including somatic cell count

Reply: Thank you for your professional advice. We first analyzed the data of production performance (milk fat, lactose and milk protein, etc.) and somatic cell number, and looked at the effect of melatonin on the prevention and treatment of mastitis in dairy cows from the apparent data, and then further explored the changes of melatonin in the blood, such as antioxidant, immune and melatonin content in the blood.

  1. The tables can be unburden from useless entries > first line “CON group” >> “group” can be deleted

Reply:We have changed the useless part of the table, in the first line“group”.

  1. Data for day 7 after cessation (d7ac) of melatonin supply > several aspects I recommend for consideration:
  2. Ithink the statistically handling of these data is not clear/transparent; the D7ac-data are not independent from the former results. Therefor they need to be incorporated into the ANOVA for repeated measurements.

Reply:Here we collected milk samples on days 0, 7, 14, 21 and 28, and blood samples on days 0, 15, and 30, at the end of the trial period. During the 7-day observation period after the end of the trial period, the last time the milk and blood samples were collected was the last day after the observation, that is the 37th day.

  1. Tab.4: Which is the rationale for these data? If there is no hypothesis for an impact of melatonin of function behind the parameters in Tab. 4, these data can be deleted or leftas supplementary material.

Reply: Based on your professional advice and after careful consideration, we have revised this section and deleted the part of Table 4.

  1. Tab. 6, title is obviously not complete

Reply:We verified the Table 6 title and have changed the title.

Modified title: Table 6 Effect of serum antioxidant indices in dairy cows after one week of melatonin cessation (n=8)

  1. Tab. related to result for d37 see above

Reply: Thank you for your professional advice. We don't understand very well this piece of advice that you mentioned. If there is any additional content in this part of the following, you can propose it, and we will make further detailed changes.

  1. 6. Discussion
  2. Comments should be added:
  3. About he recorded somatic cell count related to those which are indicating subclinical mastitis

Reply:The relationship between subclinical mastitis and somatic cell count has been added to the discussion and the economic implications of subclinical mastitis have been elaborated.The supplement is that subclinical mastitis can seriously affect milk production, according to the United States National Mastitis Board, milk production is reduced by 6% when the cow SCC is up to 50,0000, and milk production is reduced by 18% when the cow SCC is up to 1,000,000.The decline in milk production is closely related to economic performance, with studies showing a loss of US 107 for SCC between 200,000 and 400,000 per cow and US 275 for SCC above 400,000 per cow.It can be seen that the occurrence of subclinical mastitis is significantly related to somatic cell count.

  1. about the fluctuation of melatonin in blood related to season, daytime, physiological status, feed intake and performance

Reply: Based on your professional suggestions, we have made changes in the manuscript.

Revised content: The concentration of melatonin in the blood is related to the season, light, physiological state of the animal, half-life, and feed intake, with seasonal factors being the main factors affecting the concentration of melatonin in the blood.In the non-breeding season, melatonin can be used to regulate the estrus time of ewes, so that ewes can come into heat early, and in the later stage, it can reduce the mortality rate of newborns while promoting the growth and development of the fetus.In addition to seasonal factors, it is regulated by circadian rhythms, and in vivo, tryptophan, a precursor substance of melatonin, is hydroxylated to 5-hydroxytryptophan by tryptophan hydroxylase, and after decarboxylation, it becomes 5-hydroxytryptophan, serotonin increases during the day, and at night, the activity of aryl alkylamine acetyltransferase in the pineal gland increases 70-100-fold.Lastly, acetyl-5-hydroxytryptamine methyltransferase converts N-acetyl-5-hydroxytryptonin to melatonin. In this study, melatonin concentrations in the blood of dairy cows were increased by feeding exogenous melatonin.This is consistent with the findings of Shi Jianmin, which showed that the blood melatonin concentration in dairy cows can be increased by intravenous injection, subcutaneous injection, subcutaneous implantation, and oral administration of exogenous melatonin.

However, there was no significant difference in the concentration of melatonin in the blood after one week of melatonin withdrawal compared with that at the time of feeding, because the half-life of melatonin is about 30 minutes, so the effect of melatonin on the blood melatonin concentration in dairy cows after one week of melatonin withdrawal was not significant. In addition, this corresponds to the above-mentioned anti-inflammatory and antioxidant results of melatonin on dairy cow serum.

iv.about the use of melatonin and the acceptance of milk and milk products by consumers

Reply:Early, melatonin was isolated from the pineal gland of dairy cows to understand the role of melatonin in the physiological reproduction of seasonally reproducing animals.Later, melatonin was widely used to regulate the reproductive performance of mice, sheep, and dairy cows. In addition, the use of melatonin has a significant effect on the treatment of mastitis in dairy cows, by reducing the number of somatic cells, improving milk production and milk quality, and better serving people.With the improvement of people's living standards and consumption levels, more and more attention is paid to the quality of milk, which is rich in protein, fat, vitamins, lactose and mineral elements, which is close to the composition of human milk and is widely accepted by most people.Infant formula is usually cow's milk based with appropriate compounds. Therefore, the quality of milk is very important for the health of infants and adults, especially for the growth and development of infants.

References:

A R, Jumnake.; V R, Patodkar.; N E , Gavali.; V M Sardar.; P V, Mehere. Advances in Melatonin Research – A Review. Chronicle of Aquatic Science, 2024, 1, 152-174.

Is P.; Chung M.; Raman G. Breastfeeding and Maternal and Infant Health Outcomes In Develope,Evidence Reporter, 2007, 153, 1-186.

Lonnerdal, Bo. Infant formula and infant nutrition: bioactive proteins of human milk and implications for composition of infant formulas. American Journal of Clinical Nutrition, 2014, 99, 712S-717S.

iii. legal aspects for the use of melatonin

Reply:Our lab has done preliminary research on melatonin and precursors on ruminants. The legal use of melatonin has also been discussed in depth.

The effect of 5-hydroxytryptophan on milk yield and plasma hormone levels in dairy cows showed that adding 20 g/d of 5-hydroxytryptophan to dairy cow diet could increase milk yield and plasma hormone levels (PRL, GH and INS).

The effect of 5-hydroxytryptophan on the contents of serotonin and melatonin in sheep plasma showed that the addition of 50 mg/kg of 5-hydroxytryptophan to sheep diet could increase the content of melatonin and 5-hydroxytryptophan in sheep plasma to varying degrees, while the effect of serotonin was not significant.

In addition, we also refer to the research results of Liu Guoshi's team.

The Effects of Prepartum L-Tryptophan Supplementation on the Postpartum Performance of Holstein Cows showed that supplementation of Holstein cow feed with 50 mg/day or 100 mg/d L-tryptophan significantly increased blood melatonin levels.

Based on the above studies, we conducted a series of studies on the nutritional aspects of melatonin in dairy cows.

Reviewer 2 Report

Comments and Suggestions for Authors

This experiment conducted to study the effects of exogenous melatonin on performance and mastitis in dariy cows. The research is meaningful, and the experiment design was well done. 

I think the writing and data analysis should be improved.

Here are the specific suggestions.

1. Line 32-33 " the cell counts of lactating cows in the control group during melatonin feeding were significantly (p<0.05) and highly significantly (p<0.01) lower in somatic cell counts in the T120 group on days 14 and 28, respectively" 

This sentence has grammar issues, and should be improved.

2. Line 42 "milk milk component content"

There are two "milk" words.  

3. Line 107 "Figure 1 Trial design and grouping diagram"

It is better not to put the word "120 mg/d melatonin optimal" here.

4. Line 178-188

The milk sample was collected on 0, 7, 14, 21, 28, and 37, why do the SCC results only present "0, 14, 28, 37" in Figure 2?

It is better not to refer to the quadratically and linearly differences, because line chat figure cannot be displayed quadratically and linearly differences.

5. Line 199 Table 2

Line 226 Table 4

Line 241 Table 5

Line 265 Table 7

Line 294 Table 9

These tables did not have different time data, so it is not very appropriate to do the time difference analysis (Time P-value).

Comments on the Quality of English Language

Fine, should be improved.

Author Response

Draft (vetsci-3170360)

24-August-2024

Statement on the Revision of (vetsci-3170360)

Based on Reviews’ Resports

Author: Zhiqiang Cheng   Correspondence: Changjiang Zang

August 24, 2024

Reply to the Comments of Reviewer 

Reply: We thank the reviewers for comprehensive assessment of our work. We have gone through all the comments in detail and have tried our best to revise our manuscript according to the comments points by point. Additionally, revised portion are marked in red in the revised manuscript. Thanks a lot for your reviewing.

Reviewer 2

This experiment conducted to study the effects of exogenous melatonin on performance and mastitis in dariy cows. The research is meaningful, and the experiment design was well done. 

I think the writing and data analysis should be improved.

Here are the specific suggestions.

Reply: We sincerely thank the editor and all reviewers for their professional comments, which we use to improve the quality of the manuscript. The reviewers' comments are listed below in blue markers, and specific questions have been numbered. Revised portions are marked in red in the revised manuscript.

  1. Line 32-33 " the cell counts of lactating cows in the control group during melatonin feeding were significantly (p<0.05) and highly significantly (p<0.01) lower in somatic cell counts in the T120 group on days 14 and 28, respectively" 

This sentence has grammar issues, and should be improved.

Reply:We have made grammatical changes and don’t alter the content of the original text.“The results showed that the somatic cell counts of lactating cows in the CON group were lower than those in the T120 group on days 14 (p<0.05) and 28 (p<0.01) at 1 week after melatonin cessation. ”

  1. Line 42 "milk milk component content"

There are two "milk" words.  

Reply:After checking the content, we have made changes.

  1. Line 107 "Figure 1 Trial design and grouping diagram"

It is better not to put the word "120 mg/d melatonin optimal" here.

Reply: For the part of Figure 1 trial design and grouping diagram, we have made changes according to your suggestions.

  1. Line 178-188

The milk sample was collected on 0, 7, 14, 21, 28, and 37, why do the SCC results only present "0, 14, 28, 37" in Figure 2?

It is better not to refer to the quadratically and linearly differences, because line chat figure cannot be displayed quadratically and linearly differences.

Reply: Here, we collected milk samples at days 0, 7, 14, 21, and 28, and the somatic cell number was measured every 14 days, as well as the somatic cell number was measured on day 37.

We have revised the description of the results in Figure 2. The changes are as follows:

The effects of feeding melatonin and cessation feeding for one week on somatic cell counts of cows are shown in Figure 2. Compared to CON, somatic cell count of cows in T120 group on day 14 of the experiment was reduced (p<0.05), somatic cell count of cows in T120 and T160 groups on day 28 of the experiment was reduced (p<0.01), and somatic cell count of cows in T80 group was reduced (p<0.05).In addition, after one week of melatonin cessation, the somatic cell counts of cows in the trial group were lower than the CON group on day 37 of the experiment, and the differences were not significant (p>0.05).

  1. Line 199 Table 2

Line 226 Table 4

Line 241 Table 5

Line 265 Table 7

Line 294 Table 9

These tables did not have different time data, so it is not very appropriate to do the time difference analysis (Time P-value).

Reply:Here we use the MIXED program in SAS 9.4 (version 9.4, SAS Institute Inc., Cary, NC), which includes time points, fixed effects of treatments and their interactions, and random effects. The model is as follows:

Yijk=μ+trti+cowj+dayk+(trti×dayk)+eijk

Yijk is the measure of the dependent variable of the j-th cow of the i-th treatment on dayk, μ is the overall mean of the dependent variable trt, trti is the fixed effect of the i-th treatment on the dependent variable Day, k is the fixed effect of the k-day treatment on the dependent variable trt, trti×dayk is the fixed effect of the interaction between trti and dayk on the dependent variable, and eijk is the random residual effect of the j-cow in the i-th treatment on the k-th day. The data are presented as the least squares mean and the mean standard error, and p<0.05 indicates a significant difference.

Reviewer 3 Report

Comments and Suggestions for Authors

The authors' goal was to determine the impact of melatonin supplementation on somatic cell count (i.e., mastitis) in dairy cattle.  They found an effect at the second highest dose administered, which was no longer evident at a higher dose.  They also found that supplementation with melatonin had some impacts on milk components.  Overall the results are interesting, but the authors need to address a couple of methodological issues and dramatically improve their English usage.

Line

22  Here and elsewhere the authors refer to 'litter size'.  I expect the litter size for all of the cows used was one calf.  It is not clear what the authors mean.  Please clarify.

32 and throughout.  The authors use language like 'highly significantly higher (p<0.01)'.  This is redundant.  First, the statistical tests they use only identify whether means are different.  Second the p-value provides the author with information about the confidence that the means are different.  The 'significantly' or 'highly significantly' should be removed and simply state whether the value was higher or lower and provide the p-value.

52 and throughout.  There is no space after the end of the sentence.  Please correct here and throughout.

94-105.  The description of the animals and treatments needs improved.  Again, what is meant by 'litter size'?  How were the capsules administered?  It sounds as though they were put on top of the feed.  If so, how did you determine that the cows consumed all of the treatment?

104 and throughout.  Recommend you refer to the pretrial, trial and post trial.  the use of 'positive' trial is awkward and unnecessary.

138  Centrifugation should be described in g-force, not rpm.

185 and throughout.  Here you indicate that somatic cell counts were lower, but not significant.  If the means are not different (statistically), referring to them as higher or lower is inappropriate.  They are not different.

191-199.  Here and in the discussion please be clear about the fact that in many cases the highest dose was not different from control.  Why?  Please be clear about the implications.

216-225  This paragraph needs improvement.  It appears to be redundant.

Comments on the Quality of English Language

The English needs improved.  Most importantly the use of 'significantly' and 'highly significantly' should be eliminated.  The p-values that are included serve that purpose and are more meaningful for the reader.

The use of 'litter size' needs addressed.

There are some paragraphs that are internally redundant and need clarified.

Author Response

Draft (vetsci-3170360)

24-August-2024

Statement on the Revision of (vetsci-3170360)

Based on Reviews’ Resports

Author: Zhiqiang Cheng   Correspondence: Changjiang Zang

August 24, 2024

Reply to the Comments of Reviewer 

Reply: We thank the reviewers for comprehensive assessment of our work. We have gone through all the comments in detail and have tried our best to revise our manuscript according to the comments points by point. Additionally, revised portion are marked in red in the revised manuscript. Thanks a lot for your reviewing.

Reviewer 3

The authors' goal was to determine the impact of melatonin supplementation on somatic cell count (i.e., mastitis) in dairy cattle.  They found an effect at the second highest dose administered, which was no longer evident at a higher dose.  They also found that supplementation with melatonin had some impacts on milk components.  Overall the results are interesting, but the authors need to address a couple of methodological issues and dramatically improve their English usage.

22  Here and elsewhere the authors refer to 'litter size'.  I expect the litter size for all of the cows used was one calf.  It is not clear what the authors mean.  Please clarify.

Reply: The terminology used here is inappropriate. we want to express is parity, which has been revised in the manuscript.

32 and throughout.  The authors use language like 'highly significantly higher (p<0.01)'.  This is redundant. First, the statistical tests they use only identify whether means are different. Second the p-value provides the author with information about the confidence that the means are different.  The 'significantly' or 'highly significantly' should be removed and simply state whether the value was higher or lower and provide the p-value.

Reply: Based on your professional advice, we have made changes to this section in the manuscript.

52 and throughout.  There is no space after the end of the sentence.  Please correct here and throughout.

Reply: Based on your professional advice, in the manuscript we checked the formatting aspects of the manuscript and made careful revisions.

94-105.  The description of the animals and treatments needs improved.  Again, what is meant by 'litter size'?  How were the capsules administered?  It sounds as though they were put on top of the feed.  If so, how did you determine that the cows consumed all of the treatment?

Reply: Based on your professional advice,We've restated the animal and the testing method, and here we're using inappropriate terminology, and we're trying to mean parity. For the use of capsules: different levels of melatonin are put into glutinous rice capsules to feed each cow, and empty glutinous rice capsules and glutinous rice capsules containing different levels of melatonin are fed to different groups of cows before feeding, and wait for the cows to completely consume the glutinous rice capsules before ingesting the feed.

104 and throughout.  Recommend you refer to the pretrial, trial and post trial.  the use of 'positive' trial is awkward and unnecessary.

Reply:Based on your professional advice, we have made changes to the description of the trial period in the manuscript. Revised content:The trial period was 37 d, a 7 d adaptive phase was followed by 30 d of experimental period.

138  Centrifugation should be described in g-force, not rpm.

Reply: Based on your professional advice, we have revised the manuscript and carried out the conversion of units.

The radius of the centrifuge is: 6 cm

g=1.1×10-5×6×75002≈3713 g

185 and throughout. Here you indicate that somatic cell counts were lower, but not significant. If the means are not different (statistically), referring to them as higher or lower is inappropriate. They are not different.

Reply: Based on your professional suggestions, we have made changes in the manuscript. Indicators that were not statistically different have been recounted.

191-199. Here and in the discussion please be clear about the fact that in many cases the highest dose was not different from control. Why? Please be clear about the implications.

Reply:High doses of melatonin may cause negative feedback regulation, inhibit endogenous melatonin secretion, lead to an imbalance in melatonin levels in the body, affect the utilization and absorption of nutrients and immune function in dairy cows, and in addition to affecting the circadian rhythm and feeding behavior of dairy cows, these behaviors can lead to a decrease in milk production. Duan et al. showed that implanting high doses of melatonin under goat skin reduced melatonin levels in goat's muscles, kidneys, heart, and lungs. This is similar to the results of this study. In this study, high dose levels of melatonin inhibited some indicators such as milk production and milk composition. As a result, some data in the high-dose melatonin group were not different from in the control group.

References are as follows:

Tao Duan, Zi Wu ,et al.Effects of melatonin implantation on carcass characteristics, meat quality and tissue levels of melatonin and prolactin in Inner Mongolian cashmere goats[J]. Journal of Animal Science and Biotechnology, 2019, 10(04):269-276.

216-225 This paragraph needs improvement. It appears to be redundant.

Reply:Based on your professional advice and after careful consideration, we have revised this section and deleted the part of Table 4.

Round 2

Reviewer 1 Report

Comments and Suggestions for Authors

Thanks for working on the manuscript.

I recommend:

1. Tab. 2, 4, 6 and 8 present data from the period feeding melatonin. Please insert in the title of these table the day of sampling. The statistics give the information about treatmentxtime interaction. Therefore it is beneficial to have the information about sampling close by.

Please consider to combine the table 2 and 3, 4 and 5 etc. It would give a more compact data presentation - I leave this to be decided by the authors.

Data ref. to milk yield: Please simplify, e.g. "milk fat percentage (%)" can rewritten as "milk fat (%)" This would unburden the tables

Author Response

Draft (vetsci-3170360)

6-September-2024

Statement on the Revision of (vetsci-3170360)

Based on Reviews’ Resports

Author: Zhiqiang Cheng   Correspondence: Changjiang Zang

September 6, 2024

Reply to the Comments of Reviewer 

Reply: We thank the reviewers for comprehensive assessment of our work. We have gone through all the comments in detail and have tried our best to revise our manuscript according to the comments points by point. Additionally, revised portion are marked in red in the revised manuscript. Thanks a lot for your reviewing.

  1. 2, 4, 6 and 8 present data from the period feeding melatonin. Please insert in the title of these table the day of sampling. The statistics give the information about treatmentx time interaction. Therefore it is beneficial to have the information about sampling close by.

Reply: Thanks for the suggestion. The sampling time has been added to the notes in Tables 2, 4, 6, and 8 in our manuscript, and a supplement on treatment×time has been added to the statistical methods section of the manuscript.

Please consider to combine the table 2 and 3, 4 and 5 etc. It would give a more compact data presentation - I leave this to be decided by the authors.

Reply: Thanks for the suggestion. We combined Tables 2 and 3, 4 and 5, 6 and 7, and 8 and 9 in the manuscript.

Data ref. to milk yield: Please simplify, e.g. "milk fat percentage (%)" can rewritten as "milk fat (%)" This would unburden the tables

Reply: Thanks for the suggestion. We revised "milk fat percentage" and the like to "milk fat (%)" in the manuscript table.
